# Fluoride: From Nutrient to Suspected Neurotoxin

**DOI:** 10.3390/nu14173507

**Published:** 2022-08-26

**Authors:** Donald R. Taves

**Affiliations:** School of Dentistry, University of Washington, Seattle, WA 98195, USA; dtaves@pickatime.com

**Keywords:** fluoridation, fluoride wars

## Abstract

The recollections of a former public health officer and research scientist who maintained good relations with both pro- and anti-fluoridationists over the course of a 60-year career in which fluoride has gone from being a “nutrient” to a suspected neurotoxin.

## 1. Introduction

The guest editor for this special issue asked me to describe some of my many interactions over the past 60 years with those on both sides of the ongoing “war” over fluoridating drinking water [1]. Although in recent years the war has been reduced to local skirmishes over referendums, the war will undoubtedly heat up again if the federal appeals court in San Francisco rules for the plaintiff and mandates that the EPA stop fluoridation in the United States! As one of the few researchers who has maintained relationships with both pro- and anti-fluoridationists, the editor felt that my recollections would offer a unique perspective on the 60-year period in which fluoride has gone from being a “nutrient” to a suspected neurotoxin. 

## 2. Wallace Armstrong and Leon Singer

At the time when I was deciding whether to be a maverick health officer and break with the reining consensus on the safety of fluoridation, I started to question whether we could measure serum fluoride at the levels required to perform the proper renal clearance studies. I made a trip to San Francisco to meet with Armstrong and learn about the new method that he and Singer were developing for measuring serum fluoride. I was hoping that I could use their method to study the effects of renal disease on serum fluoride. Wallace, better known as Wally, was one of the few M.D.s who belonged to the American Dental Association (ADA), and the ADA’s journal published many of the early biochemical studies. I knew he was going to talk about their new method of analysis and outline the preliminary results of using it on serums from fluoridated and non-fluoridated cities at a meeting in San Francisco. I was particularly interested in their finding that there was no difference in serum fluoride due to fluoridation and at the same time finding a big difference in terms of dental caries [2]. I found that worrisome because I hypothesized that serum levels should correlate with the amount that was ingested. We exchanged our viewpoints for about an hour in the hallway outside of the meeting without convincing the other. We now know that the levels do correlate and that lack of change in their results was due to the limitations of their method for measuring serum fluoride [3] (Figure 3, p. 168, and Figure 4, p. 170).

In 1964, when I was free to pursue the fluoride analysis question, I visited Singer’s lab where they were still working on their new method. As we were talking, his lab technician interrupted to say that the values she was obtaining were still too low relative to their earlier methods. His response was to increase the length of time of diffusion. Rather than extending the time, they should have checked to see if all the fluoride had diffused. If they had done that, they would have known that they were dealing with a contaminate rather than fluoride and, with more diffusion time, they would obtain higher values due to the contamination. Over the next three years, I developed a new, more sensitive detection method using a simple silicon derivative that allowed the diffusion to work at a lower temperature and eliminated the contaminants that were giving Singer difficulty [4,5]. With these improvements, I was able to establish that there was a direct correlation between intake and serum fluoride levels and, as a result, could use this method routinely to undertake proper renal clearance studies. These improvements were incorporated into what became the standard method for measuring fluoride concentrations at low levels.

## 3. Harold Hodge and Nicholas Leone

By the end of World War 2, Harold Hodge was the leading authority on the safety of fluorides, due to his work on the atomic bomb. He became an active promoter of fluoridation based on preliminary data from the first experiments on humans, which he thought showed that fluoridation was safe and effective. He was also involved in the hearings that led to federal policy supporting fluoridation.

My relationship with Harold was complex and changed over time as he got to know me better. Initially, he only knew that I was a county health officer who had broken ranks with the consensus view of the California Department of Public Health that water fluoridation was safe. He was probably also leery of me because I questioned his assessment of the possible risk of fluoridation for people with chronic renal failure. He may have learned this from Wally or from the letter that I wrote to him in 1959 asking to study under him to acquire a PhD degree in toxicology and become a basic science researcher. At any rate, he did not respond to my letter asking to study under him. I negotiated this impasse by studying bone mineralization under Bill Neuman, the joint head of the Department of Radiation Biology. I progressed rapidly through the program and was soon appointed as an Assistant Professor in the department. Bill studied for his PhD under Hodge, but he is not otherwise mentioned here, because we never talked about fluoridation.

Soon after I arrived in 1960 to undertake my PhD with Bill, Harold’s relationship with me started to change after I went to him and asked for help in having blood samples analyzed from a woman with chronic renal failure. Within years, Harold’s and my relationship grew to the point that he—as chair of the newly created Department of Pharmacology and Toxicology—offered me an appointment in the new department, along with a promotion to Associate Professor. I was the first toxicologist in the department and able to work on whatever I pleased with ample funding, which was a great privilege. Hodge had a center grant that generously funded me and a few other members of the newly created department. My contact with him continued until he retired from his next position at UC San Francisco and he and his wife retired in Carmel, where we had our last visit.

Nicholas Leone, who performed early work on the safety of fluoridation, visited me when he was concerned about Wally being depressed, probably in the late 1970s. Wally had finally concluded that my method was superior to his and he thought that all his research had gone down the drain. I told Leone I did not agree with Wally’s conclusion. He had started the diffusion at a much lower temperature, 60 °C rather than 130 °C used in the old standard methods. Moreover, I had discovered from his work that it was possible to diffuse faster at 25 °C without the interfering contamination. That method is still the standard first step for measuring low fluoride in difficult media. At the time, I should have added that without his work on the diffusing of samples, I might not have made a good start as a researcher. I wish I had gone to visit and told him that.

## 4. John Featherstone and Howard Pollack

John and I were colleagues at the University of Rochester. He worked in the nearby Dental Research Center until he took a position at UC San Francisco in 1997, where he was eventually appointed Dean of the School of Dentistry. Our research interests overlapped and we stayed in touch over the years. He is now retired but remains quite active. John introduced me to Howard Pollock, who was his colleague at UCSF. Howard was the principal adviser on fluoridation issues for the American Dental Association. He showed me data from a California study of the caries incidence in their fluoridated and non-fluoridated water supplies. The benefits were only seen in the lower income brackets, presumably because the other brackets were using bottled rather than community water.

## 5. George W. Waldbott and John Yiamouyiannis

As noted earlier, George’s book started me thinking about patients with renal disease who logically could be expected to have some difficulty with fluoridation. I saw George Waldbott for the first time in 1977 at an AAAS Selected Symposium on the Continuing Evaluation of the Use of Fluoride. I was involved in setting up the meeting and editing the talks for publication [3]. At the meeting, George stood up, introduced himself, and commented that it was the first meeting he had attended about fluoride that was scientific. Presumably, he had attended “scientific” meetings that were only promoting fluoridation.

When I had worked out my analytic methods and we were doing them routinely, I contacted George and said that I would perform fluoride analyses for free on those of his patients whom he thought might be more sensitive to fluoridated water. He sent me a few samples and their serum fluoride levels were normal. A year or so later, we testified on opposite sides in a court case in which the plaintiff wanted to stop the water fluoridation in some municipal system. We happened to go to the men’s room at the same time and I asked him: “George, why haven’t you sent me more samples.” He replied that he just was not seeing the sensitivity reactions anymore. I thought then that there was no point in my worrying about differences in fluoride sensitivity. If George could not find them, I figured I would not be able to either, so I stopped worrying about studying the sensitive reactions.

Freeze and Lehr [1] (p. 32) describe John Yiamouyiannis as one of “the anti-fluoridation forces … most effective and charismatic spokesmen. … He was forceful but sincere in debate, and was generally liked, even by his opponents. … His written material on the other hand was shrill and alarmist.” I can attest to the authors’ appraisal; John came to visit me to talk over his claims that fluoridation caused cancer. He attended the AAAS Selected Symposium and was involved in the court case in which George and I testified. In each of those encounters, I think we exchanged brief pleasantries. In my chapter on “Claims of harm” [3], I went into considerable detail about our differences of opinion on his claims that fluoride caused an increased cancer risk. I did acknowledge, however, that “it should be remembered that while the margin of possible error can be reduced, theoretically, absolute proof of safety cannot be attained. Also, the remaining possible risks and the benefit are in different units so that a comparison will remain subjective, and science, per se, cannot make the decision.”

## 6. Albert W. Burgsthaler

Albert was the editor of the journal *Fluoride* for many years and coauthor of *Fluoridation: The Great Dilemma* [6]. I visited him at his home in Manhattan, Kansas, where he was a respected member of the chemistry department at the University of Kansas. We had many other conversations over the next 15 to 20 years, mostly at the four-day meetings of the International Society of Fluoride Research, which was held every two years in a different country. Most of the scientific sessions were on levels of fluoride intake that were higher than fluoridation, so I could only comment or ask questions. One of the three evenings of the conferences was devoted to learning about the cities we were in and having an entertaining dinner together. It is there that I became acquainted on a first-name basis with scientists opposed to fluoridation, most notably Paul Connett.

## 7. Paul and Michael Connett

Paul Connett, one of the founders of the Fluoride Action Network (FAN) and its director for 15 years, liked to have everyone singing at our conference dinners and I enjoyed assisting him. At our meeting in Beijing, China, in 2007, circumstances were such that I wound up singing a solo rendition of “I’ve Been Working on the Railroad” to Paul within earshot of at least 1000 Chinese who were waiting for a show to begin. Many of them clapped my efforts!

In 2006, I participated in a conference that Paul organized at St. Lawrence University, where he was the head of the chemistry department and a specialist in environmental chemistry and toxicology. Many in attendance did not believe that the current literature supported continuing fluoridation. I gave a short paper that reviewed new evidence that fluoride does reduce caries, which received a respectful hearing. I indicated that the usual studies that everyone relied on to argue for or against the effects of fluoridation on dental caries were based almost entirely on early teenagers, whose mature teeth would have had little time to develop, with these studies not well controlled from the point of view of evaluator bias, because the evaluator was not blinded to what group was fluoridated. The only exception at the time was a paper by Hopcraft et al. [7] that used military recruits who were older and had lived their whole lives in fluoridated or non-fluoridated communities. The radiologist who read the recruits’ dental X-rays did not know where the individuals grew up. That study showed that fluoride reduced caries by about 30%.

In 2016, FAN, along with other organizations, presented a Citizens’ Petition asking “the EPA to exercise its authority to prohibit the purposeful addition of fluoridation chemicals to U.S. water supplies on the grounds that a large body of animal, cellular, and human research showed that fluoride was neurotoxic at doses within the range now seen in fluoridated communities” (https://fluoridealert.org/issues/tsca-fluoride-trial/fact-sheet/, accessed on 5 July 2022). The court denied the EPA’s motion to dismiss the case. The ongoing TSCA Lawsuit, as it is called, relies on recognized authorities to provide considerable evidence for an association between fluoridation and lower IQ in children. The case has still not been resolved and could eventually lead to a ban on water fluoridation (for details of the case, see https://fluoridealert.org/issues/tsca-fluoride-trial/, accessed on 5 July 2022).

Paul’s son Michael, whom I met at the international meeting in Germany in 2005, is the lawyer representing the plaintiffs in the TSCA Lawsuit. Prior to becoming a lawyer, he worked with his father to develop the FAN website. He was responsible for collecting all the literature on fluoride in the world in one place. He also obtained translations of some of the early papers in Chinese and made them available on the FAN website before NIH produced their own translations (see https://fluoridealert.org/researchers/, accessed on 5 July 2022).

I have kept in touch with Michael, who continues to update me on the trial. I have not become acquainted with the experts who are testifying in the case personally, but all have experience evaluating chemical toxicity for government agencies, including the EPA. Pro-fluoridationists should not make the mistake of concluding that the anti-fluoridationists are all quacks or believers in conspiracies. This is a serious court case, and it behooves all interested parties to take the case and the evidence seriously.

## Data Availability

Not applicable.

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
