# Peer review of "Fluoride: From Nutrient to Suspected Neurotoxin"

_nutrients, 2022, doi:10.3390/nu14173507_

Round 1
Reviewer 1 Report
The author of this manuscript described the research on fluoride in drinking water based on his rich research experience, and offered a perspective on the issues considered in a court case that could end fluoridation in the U.S.
1. I think that the introduction of the relevant research personnel is too detailed and it might be better to streamline the key information.
2. Comments would be more logical and persuasive if the views of each scientist were combined with the results of the research.
3. Line 129 should be written as at a much lower temperature.
4. Line 160 should be “happened”.
5. Line 162 should be the “sensitive” reactions.
6. Line 181: should be “respected” member.
Author Response
In light of the other reviewers’ views that the commentary was highly readable, the streamlining and reorganization suggested by Reviewer #1 did not seem necessary or desirable.
Lines 160, 162, and 181 were corrected as suggested by Reviewer #1. I saw no reason to delete specific temperatures from line 129.
Reviewer 2 Report
This commentary, which is more of a personal opinion, is written in an entertaining and enjoyable manner. It provides a personal, hence quite subjective view into the struggle between the pros and cons of water-fluoridation in the past. More precise, while the cons are elaborated (potential risks for persons with renal malfunction based on the assumption of underestimation of plasma fluoride levels due to systemic errors in fluoride quantification methods not developed by the author) the pros (vast reduction of caries within the population causing an immense health benefit not only for oral, but for general health, particularly among population subgroups that are less interested in healthy nutrition, proper oral hygiene ore use of fluoridated toothpaste) are almost neglected.
Although the author criticizes lack of scientific soundness of many studies performed, dealing with negative aspects of adding fluoride to drinking water, this commentary does not provide sound or new evidence towards either side.
For me as a reviewer it is difficult to sense the journals requirements and expectations for a commentary. I have the feeling, that this commentary, as it provides mainly a glimpse into the authors personal relationships with some of the known scientific representatives dealing with fluoride research of the past decades, would better be situated elsewhere, e.g. as a teaser within a discussion group either internet based or in presence, ideally involving both, toxicologists and clinical scientist. It is not enough substantiated and may add confusion rather than meet the expectations raised by the author in the abstract and introduction section that it “…might offer a more detached perspective on the issues being considered by the federal [appeals] court [in San Francisco]” ruling “for the plaintiff and mandates that the EPA stop fluoridation in the United States”.
Author Response
In response to the reviewer’s comment that I neglected the pros of fluoridation, I would highlight the discussion of Hopcraft et al., which, as I indicated, was the first well controlled study to demonstrate the effects of drinking fluoridated water on the reduction of dental caries.
The abstract and introduction were revised to better reflect the content of the commentary.
Reviewer 3 Report
This is a very interesting and well written manuscript.
The topic is extremely current and important for readers and academics.
I would suggest only to update, check and correct references according to the Reference list and citation guide for Nutrients
Author Response
The formatting of the references has been corrected according to the Reference list and citation guide for Nutrients
Reviewer 4 Report
Dr. Taves -- I am familiar with your work and have often cited your 1974 article showing that fluoride diffuses passively across the placenta. It was a privilege to read your commentary. Its story-telling format and short vignettes made it interesting to read. I appreciated your insights and the personalized descriptions of key researchers that have played major roles in the history of the fluoridation debate. Thank you for sharing your experiences.
Author Response
reviewer #4 only had positive comments so my response is thanks.
Round 2
Reviewer 2 Report
.